# Contrast Volume-to-Estimated Glomerular Filtration Rate Ratio as a Predictor of Short-Term Outcomes Following Transcatheter Aortic Valve Implantation

**DOI:** 10.3390/jcm13102971

**Published:** 2024-05-17

**Authors:** Omar Chehab, Giulia Esposito, Edouard J. B. Long, Clarissa Ng Yin Ling, Samuel Hale, Samuel Malomo, Nanci O’Reilly, Anthony Mathur, Andreas Baumbach, Mick Ozkor, Simon Kennon, Michael Mullen

**Affiliations:** 1Department of Cardiology, Guy’s and St Thomas’ Hospitals NHS Foundation Trust, London SE1 7EH, UK; giulia.esposito3@nhs.net; 2GKT School of Medical Education, King’s College London, London SE1 1UL, UK; edouard.long@kcl.ac.uk (E.J.B.L.);; 3Department of Cardiology, Barts Health NHS Trust, London EC1A 7BE, UK; s.malomo@nhs.net (S.M.); nanci.oreilly@nhs.net (N.O.); a.baumbach@qmul.ac.uk (A.B.);

**Keywords:** acute kidney injury, contrast volume, chronic kidney disease, transcatheter aortic valve implantation, aortic stenosis

## Abstract

**Background/Objectives:** Contrast-induced acute kidney injury (AKI) is associated with early mortality and adverse events. However, in the setting of transcatheter aortic valve implantation (TAVI), previous literature has failed to establish a correlation between the absolute volume of contrast media administered and mortality. We aimed to investigate the impact of contrast volume administered normalised to estimated glomerular filtration rate (CV/eGFR) on the development of AKI and on 30-day all-cause mortality in TAVI patients. **Methods:** We retrospectively analysed a cohort of 1150 patients who underwent TAVI at our unit between 2015 and 2018. **Results:** Follow-up was complete for 1064 patients. There were 23 deaths within the follow-up period and 76 cases of AKI, 9 of which required new renal replacement therapy (RRT). Receiver-operating characteristic (ROC) curve analysis showed fair discrimination for 30-day all-cause mortality at a CV/eGFR ratio of 3.6 (area under the ROC curve (AUC) 0.671). Of patients in whom CV data were available, 86.0% (*n* = 757) had a CV/eGFR < 3.6 and 14.0% (*n* = 123) had a CV/eGFR ≥ 3.6. In multivariate logistic regression analysis, CV/eGFR ≥ 3.6 was the strongest predictor of 30-day all-cause mortality (odds ratio 5.06, 95% confidence interval [1.61–15.7], *p* = 0.004). Other independent predictors were procedural urgency (3.28 [1.04–10.3], *p* = 0.038) and being under general anaesthesia (4.81 [1.10–17.3], *p* = 0.023). CV/eGFR ≥ 3.6 was also independently associated with significantly increased odds of AKI (2.28 [1.20–4.17], *p* = 0.009) alongside significant non-left main stem coronary artery disease (2.56 [1.45–4.66], *p* = 0.001), and diabetes (1.82 [1.03–3.19], *p* = 0.037). In supplementary ROC curve analysis, a similar CV/eGFR cut point of 3.6 was found to be an excellent predictor for new RRT (AUC 0.833). **Conclusions:** In conclusion, a CV/eGFR ≥ 3.6 post-TAVI was found to be a strong predictor of 30-day mortality and AKI. The maximum contrast volume that can be safely administered in each patient without significantly increasing the risk of mortality and AKI can be calculated using this ratio.

## 1. Introduction

The number of patients treated with transcatheter aortic valve implantation (TAVI) procedures has been steadily increasing over time [1]. Notwithstanding the significant progress made in this field and the use of new-generation transcatheter heart valves (THVs), TAVI does not come without the risk of peri- and post-procedural complications, including acute kidney injury (AKI) [2]. The reported rates of AKI post-TAVI tend to vary, with studies reporting rates ranging from 6.7% to 32.5% depending on patients’ risk profiles and comorbidities [2,3,4]. More recent data from the Evolut Low Risk Trial confirmed a decreasing trend in post-procedural AKI rates amongst TAVI patients, with only 0.9% of participants developing stage 2 or 3 AKI [5]. Importantly, recent meta-analyses have found that AKI is associated not only with an almost three-fold increased risk of developing chronic kidney disease (CKD) but also with up to a four-fold higher mortality rate [6,7,8]. Given the significant long-term effects that developing AKI can have on patients’ overall health, efforts need to be made to identify subjects at higher risk to allow for increased operator awareness pre-procedurally and monitoring post-procedure.

Pre-procedural risk factors for the development of post-TAVI AKI include CKD, diabetes mellitus, peripheral vascular disease, hypertension, atrial fibrillation, and congestive heart failure; conditions which are all common in TAVI patients [3,8,9]. TAVI involves several procedural risks and characteristics that can lead to acute renal insults: intermittent periods of extreme hypotension, bleeding, cholesterol embolisation due to manipulation of catheters in atherosclerotic vessels, calcific embolisation due to THV deployment in a stenotic aortic valve and, finally, administration of contrast [4,10,11]. Contrast-induced AKI is an important risk of many cardiac procedures; however, the risk of this has been shown to be lower in TAVI than in other interventions [2]. This could to a degree be due to the immediate increase in cardiac output following TAVI, which results in improved renal perfusion and therefore improved renal function [2,11].

Recent trials comparing TAVI to surgical aortic valve replacement (SAVR) in low-risk populations have found TAVI to be non-inferior, if not superior, to SAVR [5,12]. These findings are likely to result in an expansion in the use of TAVI for the treatment of aortic stenosis (AS) to a younger patient demographic. As such, decreasing the rates of post-TAVI AKI is important to avoid the development of a complication that can result in considerable disease burden.

Of the risk factors we have mentioned, contrast volume has long been considered a potential source of renal injury, although data on this as a culprit in isolation are mixed in TAVI [8,10,11]. Studies looking exclusively at the amount of contrast volume administered without correlating it with a patient’s baseline renal function (either creatinine clearance (CrCL) or estimated glomerular filtration rate (eGFR)) have failed to show a relationship between CV administered peri-procedurally and AKI [13]. Therefore, the principal objective of this study was to investigate the relationship between the amount of contrast volume administered and the development of AKI in patients undergoing TAVI across a broad range of baseline renal functions. We also sought to assess the impact of this on short-term patient outcomes.

## 2. Methods

### 2.1. Study Population and Design

This was a single-centre retrospective cohort study that involved 1150 patients who underwent TAVI for the treatment of AS at St. Bartholomew’s Hospital (Barts Health NHS Trust) between 2015 and 2018. Ethical approval for this study was not required as it was performed as part of an outcome audit.

### 2.2. Data Collection and Study Definitions

All patients referred for consideration of TAVI underwent a series of clinical investigations including a gated computed tomography scan, electrocardiogram, and transthoracic echocardiogram. Following this, patients’ suitability for TAVI was discussed at a multi-disciplinary team meeting. Demographic, clinical, and procedural data was collected on the Trust’s database. We investigated the impact of contrast volume administered normalised to estimated glomerular filtration rate (CV/eGFR) on 30-day all-cause mortality, development of AKI, and need for renal replacement therapy (RRT) following TAVI. AKI was defined according to the AKIN classification as recommended by the Valve Academic Research Consortium-2 [14]. Most patients undergoing TAVI at our institution were referred from other hospitals and returned there for their longer-term follow-up. On that basis, we present the 30-day survival outcomes to enable the maximum number of patients to be included in the analysis.

### 2.3. Statistical Analysis

Categorical data are presented as numbers and percentages and continuous data are presented as medians and interquartile ranges. All continuous data were non-parametric based on the Shapiro–Wilk test for normality. Comparison between groups were based on chi-squared tests for categorical data and the Mann–Whitney U test for continuous data. Complete-case analysis was used in all statistical tests.

Receiver-operating characteristic (ROC) curve analysis and the Youden method were used to identify the optimal cut point for CV/eGFR to predict 30-day all-cause mortality, maximizing the sum of sensitivity and specificity. This cut-off value was used to dichotomise patients into two groups based on their CV/eGFR.

Predictors of 30-day mortality and AKI were assessed using uni- and multivariate logistic regression analysis. A *p*-value below 0.05 in univariate analysis was required for retention in the final multivariate model. To affirm the reliability and validity of the models’ results, all data were examined for multicollinearity (variance inflation factor < 5), linearity between continuous predictor variables and the logit, and influential observations. Subsequently, the area under the ROC curve (AUC) of each multivariate model was calculated to assess its performance.

In a sensitivity analysis, Firth’s penalised likelihood was applied to the logistic regression model for 30-day all-cause mortality to mitigate any bias arising from the low mortality rate. By penalizing the likelihood function, Firth’s approach effectively tackles this bias, providing more reliable parameter estimates in scenarios with limited event occurrence [15].

An additional ROC curve analysis was performed to evaluate the optimum cut-off values of CV/eGFR associated with the risk of (1) AKI and (2) new RRT.

Statistical significance was defined as *p* < 0.05. All statistical analyses were performed using R version 4.3.1 (R Foundation for Statistical Computing, Vienna, Austria) [16].

## 3. Results

During the study period, 1150 patients underwent TAVI at our unit. A total of 86 patients were lost to follow-up. The median age was 84 (79–88) years, 49.3% were female, and 21.6% (*n* = 230) of patients were urgent inpatient transfers for acute decompensated AS. The median Euroscore II was 3.01 (1.84–4.79), 24.8% had diabetes, 77.7% had hypertension, and 53.4% had a baseline eGFR < 60 mL/min/1.73 m^2^. The majority (87.8%) of implanted valves were balloon-expanding and performed through the transfemoral route (97.6%). There were 23 deaths within the follow-up period and 76 cases of AKI, 9 of which required new RRT. The peri-procedural CV administered was available for 880 patients.

ROC curve analysis showed CV/eGFR to have fair discrimination for 30-day all-cause mortality, with an AUC of 0.671 (Figure 1). The Youden method indicated an optimal cut point at a CV/eGFR of 3.6 with a specificity and sensitivity of 0.868 and 0.500, respectively.

Of patients for whom CV data were available, 86.0% (*n* = 757) had a CV/eGFR < 3.6 and 14.0% (*n* = 123) had a CV/eGFR ≥ 3.6. Table 1 compares the baseline and procedural characteristics of patients by CV/eGFR < 3.6 and ≥3.6. Patients with CV/eGFR ≥ 3.6 were more likely to have less than moderate-to-severe mitral regurgitation, a left ventricular ejection fraction <50%, higher preoperative creatinine, lower preoperative eGFR, a higher Euroscore II, had a procedural balloon aortic valvuloplasty, had received a higher volume of contrast, and had received a self-expanding valve (as opposed to a balloon-expanding valve). There were no significant differences in other baseline or procedural characteristics, including age, procedural urgency, previous myocardial infarction, revascularisation and/or surgical aortic valve replacement, significant left main stem (LMS), non-left main stem coronary disease (CAD), TAVI route of delivery, or success of valve deployment.

In univariate analysis of factors predicting 30-day all-cause mortality (Table 2), procedural urgency, higher preoperative creatinine, higher Euroscore II, and being under general anaesthesia were associated with significantly greater odds of 30-day mortality. Higher age and use of a balloon-expanding valve, as opposed to self-expanding, were associated with significantly decreased odds of 30-day mortality. Both higher CV/eGFR as a continuous variable (odds ratio (OR) 1.51, 95% confidence interval (CI): 1.12–1.95, *p* = 0.003) and CV/eGFR ≥ 3.6 as a cut-off point (OR 6.47, 95% CI: 2.18–19.2, *p* < 0.001) were associated with significantly increased odds of 30-day mortality. Notably, their CIs did not overlap. To avoid the effects of collinearity, creatinine, Euroscore II, and CV/eGFR as a continuous variable were omitted from the multivariate analysis. After multivariate analysis (Table 2), CV/eGFR ≥ 3.6 as a cut-off point was found to be the strongest predictor of 30-day all-cause mortality (OR 5.06, 95% CI: 1.61–15.7, *p* = 0.004). Other independent predictors were procedural urgency (OR 3.28, 95% CI: 1.04–10.3, *p* = 0.038) and being under general anaesthesia (OR 4.81, CI: 1.10–17.3, *p* = 0.023). The AUC-ROC of the multivariate model was 0.799, indicating its robust ability to predict 30-day postoperative mortality.

Within the group of patients where CV data was available, univariate analysis for AKI (Table 3) revealed diabetes, significant non-left main stem coronary artery disease, higher creatinine, and eGFR 30–59 mL/min/1.73 m^2^ to be factors associated with significantly increased odds of AKI. Higher AV mean gradient and AV peak gradient were associated with significantly decreased odds of developing an AKI, albeit very marginally (OR 0.98 and OR 0.99 respectively). Similarly to our 30-day mortality analysis, both CV/eGFR as a continuous variable (OR 1.27, 95% CI: 1.08–1.48, *p* = 0.003) and CV/eGFR ≥ 3.6 as a cut point (OR 2.63, 95% CI: 1.48–4.54, *p* < 0.001) were associated with significantly increased odds of developing an AKI. Creatinine, eGFR 30–59 mL/min/1.73 m^2^, and CV/eGFR as a continuous variable were not included in the multivariate analysis to avoid collinearity. In multivariate analysis (Table 3), CV/eGFR ≥ 3.6 (OR 2.28, 95% CI: 1.20–4.17, *p* = 0.009) was a strong predictor of AKI alongside significant non-left main stem coronary artery disease (OR 2.56, 95% CI: 1.45–4.66, *p* = 0.001), and diabetes (OR 1.82, 95% CI: 1.03–3.19, *p* = 0.037). With an AUC-ROC of 0.726, our multivariate model was good at distinguishing which patients were at increased odds of developing an AKI post-TAVI.

As only 14 individuals within the subset of patients with CV data available died in the follow-up period, to test the sensitivity of our findings, we repeated our multivariate analysis of 30-day all-cause mortality using Firth’s penalised logistic regression (Table 4) which helps to mitigate bias and stabilise parameter estimates in models with low event rates [15]. CV/eGFR ≥ 3.6 remained a very significant factor associated with increased odds of 30-day mortality (OR 4.79, 95% CI: 1.59–14.2, *p* = 0.006) alongside urgency (OR 3.14, 95% CI: 1.04–9.35, *p* = 0.043) and the use of general anaesthesia (OR 4.84, 95% CI: 1.19–16.4, *p* = 0.029). The AUC-ROC of the penalised model remained strong at 0.800.

An additional ROC curve analysis was performed to determine the optimum CV/eGFR cut-off point for new RRT (Figure 2) and AKI (Figure 3). In a comparison of the baseline and procedural characteristics of patients who developed an AKI compared to those who did not (Appendix A), a significantly greater proportion of AKI patients had diabetes and non-left main stem coronary artery disease. AKI patients also had significantly lower AV mean gradients, lower AV peak gradients, higher preoperative creatinine, and higher Euroscore II. There were no other significant differences in baseline characteristics between AKI and non-AKI patients. The median CV administered and eGFR was 120 mL (100–156 mL), and 52 mL/min/1.73 m^2^ (41–66 mL/min/1.73 m^2^) in AKI patients, respectively, and 117 mL (98–150 mL), and 59 mL/min/1.73 m^2^ (46–73 mL/min/1.73 m^2^) in non-AKI patients, respectively. These differences were not statistically significant. For AKI, the Youden index indicated the optimum cut-off point of CV/eGFR to be 3.3 (Table 5), with a specificity and sensitivity of 0.817 and 0.388, respectively. We also assessed the cut-off point of CV/eGFR as 3.6, which yielded a specificity and sensitivity of 0.861 and 0.300, respectively. The AUC was 0.597. For new RRT, the Youden index indicated the optimum cut-off point of CV/eGFR to be 3.6 (Table 5), similar to the cut-off point for 30-day mortality, with a specificity and sensitivity of 0.858 and 0.778, respectively. The AUC was 0.833, indicating CV/eGFR was an excellent predictor of new RRT within our sample of patients. However, interpretation of the ROC curve analysis for new RRT should be approached with caution due to the limited number of patients requiring new RRT, which may impact the generalisability and stability of the estimated predictive performance and optimum cut-off value.

## 4. Discussion

This study examined the impact of contrast volume administered normalised to eGFR on 30-day all-cause mortality and development of peri-procedural AKI. Within our cohort, 7.21% of patients developed peri-procedural AKI, which is in line with AKI rates reported by previous similar studies [2,3]. The rate of 30-day all-cause mortality was 2.16%. The main findings of our study are that:Patients with CV/eGFR ≥ 3.6 are at significantly increased odds of 30-day mortality and peri-procedural AKI.The optimal cut point for predicting peri-procedural AKI is CV/eGFR ≥ 3.3.

Our analysis revealed a CV/eGFR of 3.3 was the optimum cut-off point for the development of AKI, with an AUC of 0.597, and that CV/eGFR ≥ 3.6 was independently associated with significantly increased odds of peri-procedural AKI, with an OR of 2.28. These findings are akin to those of a study conducted by Gul et al. in a smaller cohort, which found that a CV/eGFR > 3.9 was a predictor of development of contrast-induced nephropathy [17]. Similarly, Giannini et al. found CV/eGFR > 3.2 to be the strongest predictor of AKI in their study, with an OR of 3.4 [4]. Another similar study, conducted by Venturi et al., used CrCl, rather than eGFR to investigate the association between CV/CrCl and AKI. They found that this ratio was higher in patients who developed AKI and that a CV/CrCl > 2.2 was predictive of AKI in their cohort [18].

CV/eGFR ≥ 3.6 was a strong independent predictor of 30-day all-cause mortality, with an OR of 5.06. Previous similar studies reporting on the relationship between CV/eGFR and AKI have not investigated the impact of the former on mortality. However, they reaffirmed the relationship between AKI and post-procedural mortality. Higher rates of mortality were reported in patients who developed contrast-induced nephropathy following TAVI, than in those who did not; with Gul et al. reporting 6-month mortality rates of 18.8% vs. 2.8%, and Gualano et al. reporting in-hospital mortality rates of 11.6% vs. 0.64% [3,17]. This relationship between short-term (90-day) mortality and CV administered in relation to renal function was also described by Venturi et al. However, they found that at 1 year, only AKI, not CV/CrCl, was associated with mortality [18].

Patients with CV/eGFR ≥ 3.6 were more likely to have a higher creatinine and a lower eGFR at baseline. This evidence stresses the importance of identifying patients at higher risk of developing AKI so that strategies such as pre-procedural hydration with normal saline, contrast dilution, and adjustment of contrast dose according to baseline eGFR can be put in place.

The benefits patients gain following TAVI are not only related to the cardiovascular system: by increasing renal perfusion, TAVI also reverses the pathophysiological processes of type 2 cardiorenal syndrome caused by AS, and in doing so leads to an improvement in kidney function over the 6 to 12 months following the intervention [11,19]. The development of AKI in the peri-procedural period not only invalidates this beneficial effect but also predisposes patients to the development of CKD. Future studies of a prospective nature with longer-term follow-up are recommended.

### Study Limitations

The significance of our findings is limited by the study’s retrospective nature and short-term follow-up. The study population is derived from a TAVI cohort that is older, limiting its generalizability to a younger demographic.

## 5. Conclusions

Few studies have analysed the direct relationship between CV/eGFR and mortality in patients undergoing TAVI. In our analysis, we have shown that contrast volume in isolation is not predictive of significant clinical events, whereas patients with CV/eGFR ≥ 3.6 have over double the odds of peri-procedural AKI and over five-fold the odds of 30-day all-cause mortality. There is a degree of agreement between the findings of this study and the available literature suggesting a potential utility of CV/eGFR when risk-stratifying patients for AKI prior to and following TAVI.

## Figures and Tables

**Figure 1 jcm-13-02971-f001:**
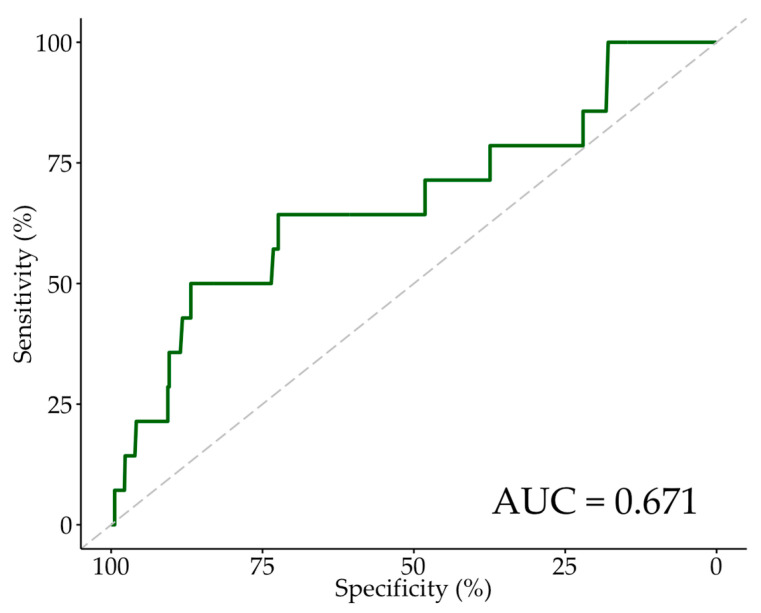
ROC curve for 30-day mortality. Abbreviations: ROC—receiver-operating characteristic, AUC—area under the ROC curve.

**Figure 2 jcm-13-02971-f002:**
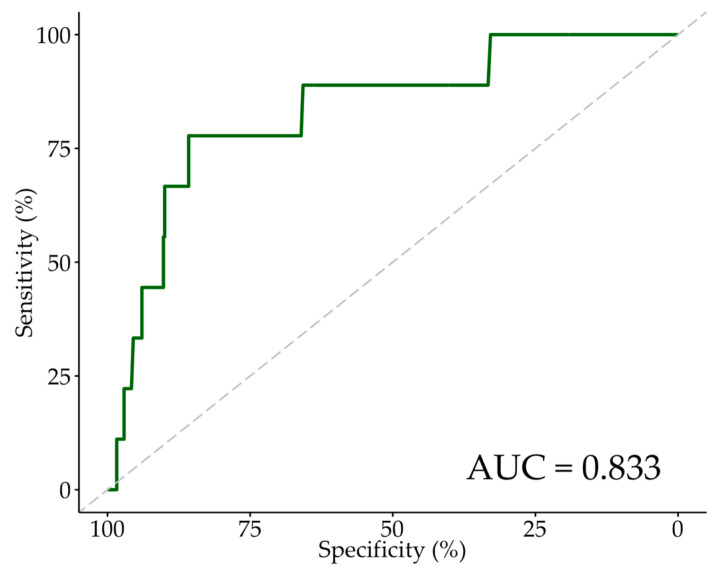
ROC curve for new RRT. Abbreviations: ROC—receiver-operating characteristic, RRT—renal replacement therapy, AUC—area under the ROC curve.

**Figure 3 jcm-13-02971-f003:**
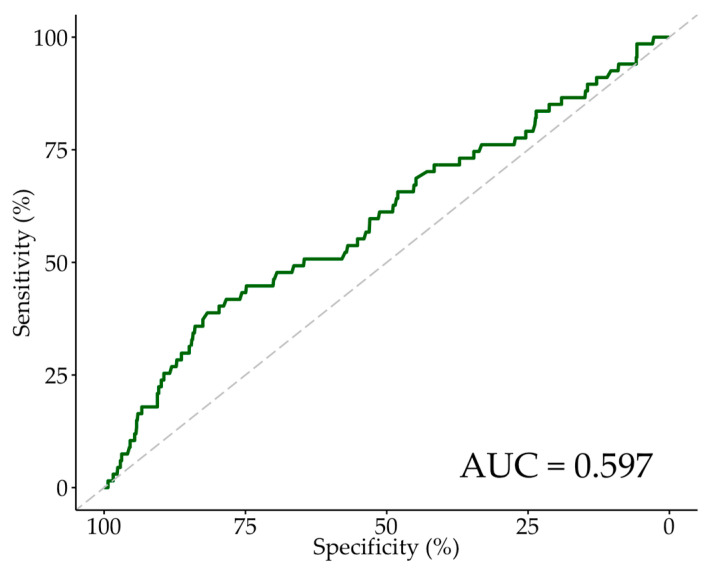
ROC Curve for AKI. Abbreviations: ROC—receiver-operating characteristic, AKI—acute kidney injury, AUC—area under the ROC curve.

**Table 1 jcm-13-02971-t001:** Baseline and procedural characteristics of study participants by CV/eGFR.

	CV/eGFR < 3.6 (*n* = 757)	CV/eGFR ≥ 3.6 (*n* = 123)	* *p*-Value
Age, years	84 (79–88)	83 (79–87)	0.471
Female	377 (50)	61 (50)	0.966
Urgent	170 (22)	28 (23)	0.940
Diabetes	190 (25)	37 (30)	0.241
Hypertension	590 (78)	103 (84)	0.145
Previous MI	123 (16)	23 (19)	0.498
Prior revascularisation	112 (15)	25 (20)	0.117
SAVR history	15 (2.0)	3 (2.5)	0.728
Previous stroke/TIA	130 (17)	22 (18)	0.846
PVD	51 (6.7)	14 (11)	0.068
Chronic pulmonary disease	172 (23)	34 (28)	0.232
Indication (AS or AR), AS	791 (98.8)	339 (97.13)	0.054
NYHA ≥ 3	459 (61)	80 (65)	0.370
Significant LMS disease	56 (7.8)	10 (8.5)	0.811
Significant non-LMS CAD	295 (42)	53 (45)	0.493
≥Moderate MR	511 (70)	73 (61)	0.036
LVEF, %			<0.001
≥50	160 (21)	18 (15)	
30–50	134 (18)	42 (34)	
<30	458 (61)	63 (51)	
AV mean gradient, mmHg	42 (35–51)	42 (32–50)	0.279
AV peak gradient, mmHg	71 (60–85)	70 (60–82)	0.476
AV area, cm^2^	0.70 (0.60–0.80)	0.70 (0.60–0.80)	0.925
PA systolic pressure, mmHg	36 (30–47)	36 (30–48)	0.842
Creatinine, μmol/L	91 (75–111)	133 (106–180)	<0.001
eGFR, mL/min/1.73 m^2^	60 (50–74)	39 (30–49)	<0.001
eGFR, mL/min/1.73 m^2^			<0.001
≥60	404 (53)	10 (8.1)	
30–59	337 (45)	82 (67)	
<30	16 (2.1)	31 (25)	
Euroscore II	2.95 (1.82–4.65)	3.85 (2.70–7.38)	<0.001
GA	38 (5.0)	11 (8.9)	0.078
Procedural BAV	105 (14)	27 (22)	0.020
Contrast volume used, mL	110 (94–140)	180 (134–233)	<0.001
TAVI delivery, femoral	737 (98)	120 (98)	0.752
Balloon-expanding valve	672 (89)	98 (80)	0.003
CV/eGFR	2.00 (1.50–2.54)	4.54 (4.02–5.26)	<0.001
Successful valve deployment	722 (96)	114 (93)	0.179

Abbreviations: CV/eGFR—contrast volume administered normalised to estimated glomerular filtration rate, MI—myocardial infarction, SAVR—surgical aortic valve replacement, TIA—transient ischaemic attack, PVD—peripheral vascular disease, AS—aortic stenosis, AR—aortic regurgitation, NYHA—New York Heart Association, LMS—left main stem, CAD—coronary artery disease, MR—mitral regurgitation, LVEF—left ventricular ejection fraction, AV—aortic valve, PA—pulmonary artery, eGFR—estimated glomerular filtration rate, GA—general anaesthesia, BAV—bicuspid aortic valve, TAVI—transcatheter aortic valve implantation. Data presented as medians (interquartile range) or number (percentage). * *p*-value was based on chi-squared or Mann–Whitney U test.

**Table 2 jcm-13-02971-t002:** Univariate and multivariate association for 30-day all-cause mortality.

	Univariate		Multivariate	
	OR (95% CI)	*p*-Value	OR (95% CI)	*p*-Value
Age, years	0.95 (0.91–0.99)	0.019	0.99 (0.93–1.07)	0.771
Female	0.74 (0.32–1.71)	0.488		
Urgency	4.12 (1.78–9.61)	<0.001	3.28 (1.04–10.3)	0.038
Diabetes	1.64 (0.65–3.81)	0.268		
Hypertension	0.65 (0.27–1.70)	0.345		
Previous MI	1.09 (0.31–2.93)	0.882		
Prior revascularisation	1.10 (0.32–2.98)	0.862		
Prior SAVR	1.84 (0.10–9.34)	0.557		
PVD	1.93 (0.45, 5.80)	0.296		
Chronic pulmonary disease	1.39 (0.53–3.30)	0.472		
NHYA ≥ 3	2.24 (0.89–6.82)	0.114		
Significant LMS disease	0.53 (0.03–2.58)	0.539		
Significant non-LMS CAD	1.03 (0.43–2.36)	0.950		
≥Moderate MR	1.42 (0.59–3.28)	0.416		
LVEF, %				
>50	0.88 (0.28–2.69)	0.818		
30–50	0.48 (0.18–1.35)	0.146		
<30	-			
AV mean gradient, mmHg	0.97 (0.94–1.00)	0.082		
AV peak gradient, mmHg	0.98 (0.96, 1.00)	0.109		
AV area, cm^2^	0.99 (0.99–0.99)	0.884		
PA systolic pressure, mmHg	1.03 (1.00–1.06)	0.066		
Creatinine, μmol/L	1.01 (1.00–1.01)	0.046 *		
eGFR, mL/min/1.73 m^2^				
≥60	-			
30–60	1.22 (0.50–3.06)	0.658		
<30	3.37 (0.73–11.7)	0.075		
Euroscore II	1.09 (1.04–1.14)	<0.001 *		
GA	9.02 (3.51–21.7)	<0.001	4.81 (1.10–17.3)	0.023
Procedural BAV	0.83 (0.19–2.47)	0.772		
Contrast volume used, mL	1.01 (1.00–1.01)	0.105		
CV/eGFR	1.51 (1.12–1.95)	0.003 *		
CV/eGFR ≥ 3.6	6.47 (2.18–19.2)	<0.001	5.06 (1.61–15.7)	0.004
TAVI access, femoral	4.19 (0.65–15.5)	0.062		
Balloon-expanding valve	0.38 (0.16–1.08)	0.047	0.48 (0.14–1.94)	0.255

Abbreviations: OR—odds ratio, CI—confidence interval, MI—myocardial infarction, SAVR—surgical aortic valve replacement, PVD—peripheral vascular disease, LMS—left main stem, CAD—coronary artery disease, MR—mitral regurgitation, LVEF—left ventricular ejection fraction, AV—aortic valve, PA—pulmonary artery, eGFR—estimated glomerular function rate, GA—general anaesthesia, BAV—bicuspid aortic valve, CV/eGFR—contrast volume administered normalised to estimated glomerular filtration rate, TAVI—transcatheter aortic valve implantation, AUC-ROC—area under the receiver-operating characteristic curve. * Omitted in multivariate model to avoid collinearity. AUC-ROC of multivariate model: 0.799.

**Table 3 jcm-13-02971-t003:** Univariate and multivariate association for AKI.

	Univariate		Multivariate	
	OR (95% CI)	*p*-Value	OR (95% CI)	*p*-Value
Age, years	0.99 (0.96–1.02)	0.526		
Gender, females	1.39 (0.89–2.18)	0.148		
Urgency	1.06 (0.61–1.75)	0.837		
Diabetes	2.03 (1.28–3.18)	0.002	1.82 (1.03–3.19)	0.037
Hypertension	1.03 (0.62–1.80)	0.901		
Previous MI	1.67 (0.96–2.78)	0.057		
Prior revascularisation	1.22 (0.67–2.11)	0.488		
Prior SAVR	0.46 (0.03–2.23)	0.454		
PVD	1.72 (0.81–3.32)	0.130		
Chronic pulmonary disease	1.23 (0.74–1.98)	0.418		
NHYA ≥ 3	1.22 (0.77–1.98)	0.399		
Significant LMS disease	1.09 (0.45–2.30)	0.828		
Significant non-LMS CAD	2.08 (1.31–3.34)	0.002	2.56 (1.45–4.66)	0.001
≥Moderate MR	1.00 (0.61–1.61)	0.997		
LVEF, %				
>50	0.90 (0.48–1.68)	0.741		
30–50	0.67 (0.39–1.18)	0.152		
<30	-			
AV mean gradient, mmHg	0.98 (0.96–0.99)	0.008	0.99 (0.95–1.03)	0.626
AV peak gradient, mmHg	0.99 (0.98–1.00)	0.007	0.99 (0.97–1.02)	0.370
AV area, cm^2^	0.98 (0.98–0.98)	0.775		
PA systolic pressure, mmHg	1.01 (0.99–1.03)	0.224		
Creatinine, μmol/L	1.01 (1.00–1.01)	0.001 *		
eGFR, mL/min/1.73 m^2^				
≥60	-			
30–59	1.72 (1.08–2.79)	0.025 *		
<30	1.90 (0.74–4.29)	0.147		
Euroscore II	1.03 (0.98–1.07)	0.191		
GA	1.97 (0.88–3.93)	0.073		
Procedural BAV	0.96 (0.50–1.72)	0.907		
Contrast volume used, mL	1.00 (1.00–1.01)	0.377		
CV/eGFR	1.27 (1.08–1.48)	0.003 *		
CV/eGFR ≥ 3.6	2.63 (1.48–4.54)	<0.001	2.28 (1.20–4.17)	0.009
TAVI access, femoral	2.18 (0.63–5.82)	0.160		
Balloon-expanding valve	0.76 (0.42–1.47)	0.380		

Abbreviations: AKI—acute kidney injury, OR—odds ratio, CI—confidence interval, MI—myocardial infarction, SAVR—surgical aortic valve replacement, PVD—peripheral vascular disease, LMS—left main stem, CAD—coronary artery disease, MR—mitral regurgitation, LVEF—left ventricular ejection fraction, AV—aortic valve, PA—pulmonary artery, eGFR—estimated glomerular function rate, GA—general anaesthesia, BAV—bicuspid aortic valve, CV/eGFR—contrast volume administered normalised to estimated glomerular filtration rate, TAVI—transcatheter aortic valve implantation, AUC-ROC—area under the receiver-operating characteristic curve. * Omitted in multivariate model to avoid collinearity. AUC-ROC of multivariate model: 0.726.

**Table 4 jcm-13-02971-t004:** Multivariate association for 30-day all-cause mortality with CV/eGFR ≥ 3.6 using Firth’s penalised likelihood.

	OR (95% CI)	*p*-Value
Age, years	0.99 (0.93–1.06)	0.719
Urgency	3.14 (1.04–9.35)	0.043
GA	4.84 (1.19–16.4)	0.029
CV/eGFR ≥ 3.6	4.79 (1.59–14.2)	0.006
Balloon-expanding valve	0.46 (0.14–1.72)	0.229

Abbreviations: OR—odds ratio, CI—confidence interval, GA—general anaesthesia, CV/eGFR—contrast volume administered normalised to estimated glomerular filtration rate, AUC-ROC—area under the receiver-operating characteristic curve. AUC-ROC: 0.800.

**Table 5 jcm-13-02971-t005:** Summary of cut-off values from ROC curves.

	Sensitivity	Specificity	NPV	PPV
30-day Mortality, AUC: 0.671				
CV/eGFR ≥ 3.6 (Youden Index)	0.500	0.868	0.991	0.058
AKI, AUC: 0.597				
CV/eGFR ≥ 3.6	0.300	0.861	0.941	0.142
CV/eGFR ≥ 3.3 (Youden Index)	0.388	0.817	0.945	0.145
New RRT, AUC: 0.833				
CV/eGFR ≥ 3.6 (Youden Index)	0.778	0.858	0.997	0.050

Abbreviations: ROC—receiver-operating characteristic, NPV—negative predictive value, PPV—positive predictive value, AUC—area under the ROC curve, CV/eGFR—contrast volume administered normalised to estimated glomerular filtration rate, AKI—acute kidney injury, RRT—renal replacement therapy.

## Data Availability

The data presented in this study are available on request from the corresponding author due to privacy reasons.

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
