# Peer review of "Contrast Volume-to-Estimated Glomerular Filtration Rate Ratio as a Predictor of Short-Term Outcomes Following Transcatheter Aortic Valve Implantation"

_jcm, 2024, doi:10.3390/jcm13102971_

Round 1

Reviewer 1 Report

Comments and Suggestions for Authors

Article is well written and deals with an interesting topic. Few issues should be revised:

- references should be updated as many recent paper have been published on this topic; discussion should be revised accordingly

- please include a table describing sensitivity / specificity / PPV / NPV for each cut-off

- 30-days mortality is rarely considered a time-to-event endpoint in cardiovascular literature. Lines 144-147 are confusing ("equality of survivor functions" with P<0.001 but CI overlapping, even if not shown in figure...). Please remove Kaplan-Meier analysis and replace with actuarial analysis (chi square). Change discussion accordingly.

- avoid text-table duplication of results (e.g. regression analysis)

- when using Cox regression, proportional hazard assumption should be checked and criteria should be met. I suggest removing time-to-event analysis from this paper to strengthen results and reduce any bias.

Reviewer 2 Report

Comments and Suggestions for Authors

This current manuscript presents a retrospective study that explores the relationship between contrast volume-to-estimated glomerular filtration rate (CV/eGFR) and acute kidney injury (AKI) in patients undergoing transcatheter aortic valve implantation (TAVI). Especially, tt identifies a CV/eGFR ≥3.6 post-TAVI as a strong predictor of 30-day mortality, providing valuable clinical insights for renal risk management in TAVI procedures. Overall, the study is clear and well-structured, presenting solid clinical sound. However, future studies are recommended that include a prospective approach with longer follow-up periods to confirm and expand the findings here.

Nevertheless, some areas require attention:

1.       To investigate the relationship between contrast volume and AKI development in TAVI patients with varying baseline renal functions, it is advisable to compare characteristics between the AKI and non-AKI groups, including the types and doses of contrast used.

2.       Given that CKD is a significant risk factor for AKI and that 53.7% of patients had a baseline eGFR <60 ml/min/1.73m2, incorporating CKD history into the analysis in Tables 1 and 2 would be beneficial.

3.       The study should assess whether different stages of AKI contribute to mortality differences.

4.       With only 8 new RRT patients, strategies to mitigate result bias should be discussed.

5.       Longer follow-up periods would enhance the study's validity.

6.       A more thorough literature review is advised to contextualize the discussion.

7.       Future studies should consider a prospective design with extended follow-up durations
